# Recent Advancements in Studies on Chemosensory Mechanisms Underlying Detection of Semiochemicals in Dacini Fruit Flies of Economic Importance (Diptera: Tephritidae)

**DOI:** 10.3390/insects12020106

**Published:** 2021-01-26

**Authors:** Hajime Ono, Alvin Kah-Wei Hee, Hongbo Jiang

**Affiliations:** 1Graduate School of Agriculture, Kyoto University, Kyoto 606-8502, Japan; 2Department of Biology, Faculty of Science, Universiti Putra Malaysia, 43400 UPM Serdang, Selangor Darul Ehsan, Malaysia; alvinhee@upm.edu.my; 3Key Laboratory of Entomology and Pest Control Engineering, College of Plant Protection, Southwest University, Chongqing 400716, China; jhb8342@swu.edu.cn; 4Academy of Agricultural Sciences, Southwest University, Chongqing 400716, China

**Keywords:** Dacini fruit fly, chemosensory, semiochemical, male attractant, sex pheromone, chemosensory receptor, olfactory receptor, *Bactrocera*, *Zeugodacus*

## Abstract

**Simple Summary:**

Chemical information, among various environmental cues, is the most important factor for insect survival and reproduction. Insects usually rely on pheromone signals, including sex, aggregation, and alarm pheromones, in conspecific communications. For phytophagous insects, plant semiochemicals associated with insect–plant interactions profoundly affect insect behavior and physiology. Therefore, elucidation of the mechanisms underlying the perception of various chemical signals is essential to understand the adaptation of insects to their surrounding environment. Dacini fruit flies are interesting models to investigate chemosensory mechanisms because they utilize various chemical cues, such as floral fragrances in mutualistic relationships, volatiles released from host fruits, and sex pheromones in mating behaviors. Since many Dacini species are serious pests, novel insights into chemosensory mechanisms not only contribute to understanding basic principles of chemoreception, but also provide cues for the development of more effective agents for pest control. Herein, we review current knowledge on chemosensory mechanisms and related topics in Dacini fruit flies from various aspects of chemical ecology, physiology, neuroscience, and molecular biology. We also discuss future perspectives based on recent advancements in these studies.

**Abstract:**

Dacini fruit flies mainly contain two genera, *Bactrocera* and *Zeugodacus*, and include many important pests of fruits and vegetables. Their life cycle is affected by various environmental cues. Among them, multiple characteristic semiochemicals have remarkable effects on their reproductive and host-finding behaviors. Notably, floral fragrances released from so-called fruit fly orchids strongly attract males of several Dacini fruit fly species. Focusing on the strong attraction of male flies to particular chemicals, natural and synthetic lures have been used for pest management. Thus, the perception of semiochemicals is important to understand environmental adaptation in Dacini fruit flies. Since next-generation sequencers are available, a large number of chemosensory-related genes have been identified in Dacini fruit flies, as well as other insects. Furthermore, recent studies have succeeded in the functional analyses of olfactory receptors in response to semiochemicals. Thus, characterization of molecular components required for chemoreception is under way. However, the mechanisms underlying chemoreception remain largely unknown. This paper reviews recent findings on peripheral mechanisms in the perception of odors in Dacini fruit flies, describing related studies in other dipteran species, mainly the model insect *Drosophila*
*melanogaster*. Based on the review, important themes for future research have also been discussed.

## 1. Introduction

Tephritid fruit flies (Diptera: Tephritidae) contain many notorious pests of commercially important fruits and vegetables. Females lay their eggs directly into fruits, and subsequently hatched larvae feed and develop in the fruits. Therefore, the consequent damage causes heavy economic losses by rendering infested fruits inedible and prohibiting fruit exports due to strict quarantine restrictions. Among the tephritid fruit flies, the Tribe Dacini is one of the most species-rich clades with more than 700 species, including three crucial genera: *Bactrocera*, *Zeugodacus*, and *Dacus* [1]. Dacini fruit flies have been extensively studied in many aspects such as their ecology, behavior, and physiology. In general, phytophagous insects respond to environmental cues, including the olfactory and visual elements associated with their host plants [2]. Among various environmental information, plant semiochemicals play crucial roles in the life cycles of Dacini fruit flies. For example, male Oriental fruit flies, *Bactrocera dorsalis*, are strongly attracted to a specific phenylpropanoid, methyl eugenol (ME), which is an element of floral fragrance present in “fruit fly orchids” in the genus *Bulbophyllum* (Orchidaceae) [3]. ME is found in various plant organs of at least 450 species from 80 families [3]. Due to its strong attraction, ME has been widely used as a lure in pest control programs against *B*. *dorsalis*. It is noteworthy that the use of ME for male annihilation successfully led to the eradication of *B*. *dorsalis* from entire islands in the Marianas and Okinawa located in the North Pacific Ocean [4,5,6]. In addition to the aromatic floral component, volatiles derived from host fruits play a crucial role in the search for oviposition sites by gravid females of Dacini fruit flies. Electroantennogram (EAG) and gas chromatography-flame ionization detection coupled with electroantennographic detection (GC-EAD) studies have identified various volatiles released from host fruits that are detected by the antennae of *B*. *dorsalis* females [7,8,9,10,11]. The active compounds mainly contained terpenes and phenylpropanoids. Moreover, several attractive compounds have been identified through behavioral tests using olfactometers [7,8,9]. Although the perception of various semiochemicals is very important in the diverse life histories of Dacini fruit flies, the mechanisms by which chemoreception occurs have not been fully elucidated. In this review, we highlight recent advancements in the elucidation of chemosensory mechanisms of Dacini fruit flies.

## 2. Semiochemicals Involved in the Life Cycle of Dacini Fruit Flies

### 2.1. Male Attractants and Sex Pheromones

Both male attractants and sex pheromones have been extensively characterized in Dacini fruit flies (Figure 1). As there are excellent reviews [12,13,14], we briefly describe major discoveries in Dacini male attractants and sex pheromones. The existence of male lures such as ME was first discovered approximately a hundred years ago [15,16]. Since then, many male lures, e.g., isoeugenol analogs for *Z*. *diversus*, have been identified [16,17]. In particular, males of many *Bactrocera* and *Zeugodacus* species are strongly attracted to certain floral fragrances, either ME, or raspberry ketone (RK: a phenylbutanoid) [18]. Based on male attractiveness, Dacini flies can be classified into two groups: ME- and RK-sensitive species [19]. Following the attraction, males voraciously ingest the attractants [20,21]. Notably, the synthetic attractant, cue-lure, which is analogous to the acetyl derivative of RK, has been developed and used for pest management of RK-sensitive fruit fly species, particularly the melon fly *Zeugodacus cucurbitae* [22]. The robust attractiveness and feeding behavior has been thought to partially enhance mating success by the acquisition of male sex pheromones or precursors to attract conspecific females [13]. Furthermore, zingerone (a phenylbutanoid) has been characterized as an attractant for both ME- and RK-sensitive fruit fly species, likely owing to the hybrid chemical structure between ME and RK [23]. ME-sensitive fruit flies convert the ingested ME into other phenylpropanoids, such as (*E*)-coniferyl alcohol (E-CF) and 2-allyl-4,5-dimethoxyphenol (DMP), as male sex pheromones [24]. The converted compounds accumulate in the rectal gland, which is considered to be a secretory organ of male sex pheromones [25,26]. RK-sensitive fruit flies accumulate ingested RK without conversion into rectal glands [27,28]. Thus, the robust attractiveness of male attractants seems to be intimately associated with male sex pheromone production and accumulation.

Attraction of female to male sex pheromones has been well studied in the ME-sensitive fruit fly *B*. *dorsalis*. Females were found to be attracted not only to live conspecific males, but also to male rectal glands in laboratory assays [29]. Of the two sex pheromones, E-CF triggered more pheromone-mediated zigzag behavior in females than DMP, although mixing of DMP and E-CF enhanced this behavior in a wind tunnel bioassay [30,31]. Unlike the male attractant ME, the male sex pheromones released from the rectal glands as smoke could function at close range rather than long range for attraction [29,32]. Light intensity greatly affects the female response, as females are most sensitive to both sex pheromones under light intensity corresponding to that at dusk [33]. For the RK-sensitive fruit fly, *Z*. *cucurbitae*, females were attracted to live conspecific males, but did not respond to male rectal glands [29]. Females were attracted more to males fed with cue-lure or zingerone than to unfed males [33]. However, to the best of our knowledge, an obvious pheromonal activity of the RK-related phenylbutanoids has not been reported [13,14].

Unlike the production of sex pheromones by males of ME- and RK-sensitive fruit flies, females released sex pheromones to attract males in the olive fly *B*. *oleae* [34]. A major component of the sex pheromones has been identified as the spiroketal 1,7-dioxaspiro[5,5]undecane, more commonly known as olean [35,36]. There are the (*R*)- and (*S*)-enantiomers in olean, and female flies release it as a racemic mixture. Remarkably, males responded only to (*R*)-olean, which functions as a sex pheromone. In contrast, females responded only to (*S*)-olean in a short range, probably owing to its aphrodisiac action [37]. As various spiroketals are commonly present in the rectal glands of Dacini fruit flies [38], a process to acquire the sexually dimorphic response to the enantiomers in *B*. *oleae* is interesting in terms of its evolutionary and physiological aspects.

Since the success of effective pest management of RK-sensitive fruit flies using cue-lure [22], several lures have been developed as practical agents for monitoring pest fruit flies [13]. Perception of these attractants and male sex pheromones by the peripheral sensory organs, as well as the processing of this information in the central nervous system, are crucial themes, as described below.

### 2.2. Common Plant Volatiles Other Than Phenylpropanoids and Phenylbutanoids as Attractants

Common plant volatiles released from host or non-host plants are also important for the reproductive and host-finding behaviors of Dacini fruit flies [39]. Electrophysiological studies using EAG and GC-EAD have been used to identify multiple volatiles released from host fruits that are detected in the female antennae of several fruit flies. For example, multiple compounds, including aliphatic alcohols, aliphatic aldehydes, esters, terpenes, and phenolic compounds, have been identified as antennal responsive compounds of *B*. *dorsalis* from host fruits, such as tropical almond fruit (*Terminalia catappa*) and mango (*Mangifera indica*), by extensive electrophysiological studies [7,8,9,10,40]. Furthermore, some of the volatiles exhibited attractiveness of female flies in the olfactometer tests [8,9]. In addition, several volatiles from mango elicited oviposition behavior [10,41]. The Chinese citrus fruit fly *B*. *minax* is a univoltine and oligophagous species as its host range is restricted to citrus fruits. Female *B*. *minax* was attracted to four volatile components, namely linalool, nonanal, citral, and limonene, contained in citrus peels among the 13 compounds tested by Y-tube olfactometer tests. In contrast, furyl alcohol and guaiacol exhibited deterrent activities against females [42]. The attractiveness of analogous monoterpenes to females is an interesting aspect of chemoreception associated with host-finding and oviposition behaviors in citrus fruit flies.

Among the common plant volatiles, sesquiterpenoids, including β-caryophyllene and α-humulene, along with ME-derived phenylpropanoids, were sequestered in male rectal glands of the ME-sensitive guava fruit fly *B*. *correcta* [43]. A field trap test at a guava orchard in Thailand revealed that a mixture of β-caryophyllene and α-humulene attracted only a few males, in contrast to the significant attractiveness of ME [43]. In another field trap test conducted in Thailand, using β-caryophyllene resulted in the capture of seven times more *B*. *correcta* males than ME-baited traps [44]. While ME attracted only mature males, β-caryophyllene could also attract immature males, suggesting an effective lure candidate [44,45]. The attractiveness of female *B*. *correcta* and *B*. *dorsalis* to β-caryophyllene at high concentrations has also been reported using a Y-tube olfactometer [46].

Remarkably, α-ionone analogs (norsesquiterpenes) are attractive to *B*. *latifrons* males, although this species does not respond to typical aromatic attractants such as ME, RK, and cue-lure. First, α-ionol analogs present in essential oils have been identified as potential male attractants in laboratory behavioral bioassays [47]. Furthermore, 3-oxo-α-ionol analogs exhibited stronger attractiveness as well as phagostimulant activity to *B*. *latifrons* [48]. Interestingly, males of *B*. *latifrons* ingested 3-hydroxy-α-ionone, present in eggplant as a minor component, and subsequently accumulated it and the 3-oxidized derivatives in their rectal glands [49]. The accumulation reminds us the transformation of ingested ME into male sex pheromones in *B*. *dorsalis*. However, 3-oxo-α-ionol analogs have not been characterized as natural male attractants and sex pheromones. Based on this information, effective male attractants such as 3-oxo-7,8-dihydro-α-ionone modified from 3-oxo-α-ionol have been developed [50]. As *B*. *latifrons* and *B*. *dorsalis* are closely related species in phylogenetic analyses [1], the differential attraction of these species to ME or 3-oxo-α-ionone, respectively, is intriguing.

## 3. Detection of Male Attractants by the Peripheral Sensory Organs

The detection of male attractants in Dacini fruit flies is an important issue for understanding their chemosensory mechanisms. Flies are equipped with two peripheral olfactory organs: the antennae and maxillary palps [51]. In general, the antennae mainly detect pheromones and various semiochemicals as the primary olfactory organs. In contrast, the role of the maxillary palps has not been well characterized in many insects. Identifying the organ involved in the detection of male attractants is of much interest. Notably, recent studies have demonstrated that male attractants are detected not only by the antennae, but also by the maxillary palps, entailing different effects among fruit fly species [52,53,54]. For ME-sensitive *B*. *dorsalis*, ablation of male antennae remarkably reduced responsiveness to ME at a short range [52]. Furthermore, males lost the ability to detect ME at a long distance. In contrast, ablation of male maxillary palps did not cause a loss of ME-detection, although they took longer to detect ME. Therefore, the antennae and maxillary palps seem to play different roles in the detection of ME, depending on the distance from the lure source. Interestingly, the effects of ablation of the chemosensory organs in the Queensland fruit fly *B*. *tryoni* were different from those in *B*. *dorsalis* [53]. Males of *B*. *tryoni* are strongly attracted to RK and cue-lure. Behavioral experiments have shown that the preference for cue-lure in a short range was abolished by the ablation of maxillary palps, but not the antennae, which is different from the case in *B*. *dorsalis*. Furthermore, electrophysiological analyses in RK-sensitive fruit flies have provided valuable insights into the recognition of attractants by maxillary palps. The maxillary palps of *B*. *tryoni* responded strikingly to cue-lure, whereas only a weak response was observed in the antennae [53]. Likewise, the maxillary palps also exhibited remarkable electrophysiological responses to cue-lure as well as RK in another RK-sensitive striped fruit fly, *Z*. *scutellata* [54]. The maxillary palps of both males and females responded to cue-lure in *B*. *tryoni* and *Z*. *scutellata*, suggesting that the male-specific attraction is triggered by sexually dimorphic processing of chemosensory information in the central nervous system, rather than in the peripheral tissues, in these species. It should be noted that the association of the maxillary palps with gustatory stimulation has been reported in other dipteran species. Detection of pleasant odors via the maxillary palps enhanced feeding behaviors in *D*. *melanogaster* and the blow fly *Phormia regina* [55,56]. Furthermore, axonal projection of olfactory receptor neurons from the maxillary palps to the primary gustatory center, the subesophageal ganglion, was observed using an anterograde staining method [56]. Considering that the attraction to male lures triggers compulsive feeding behavior in Dacini fruit flies, this raises the possibility that the maxillary palps convey olfactory information to the primary gustatory center to induce the gustatory response.

It is possible that gustatory organs, such as the proboscis and tarsi, are involved in the detection of aromatic attractants. To confirm this, electrophysiological analyses, such as the tip-recording method, would be useful [57]. However, to the best of our knowledge, there is no report on the response of the gustatory sensilla to male attractants. Analysis of gustatory responses to male attractants at the level of sensilla will be necessary to understand the mechanisms underlying the feeding behavior triggered by olfactory perception in Dacini fruit flies.

## 4. Chemosensory-Related Behaviors Modulated by Physiological Status

The response thresholds of insects to external chemical and visual stimuli are modulated by their physiological status [58,59]. Therefore, an understanding of the daily rhythm, sexual maturation, mating, and feeding status of flies is important in undertaking Dacini chemosensory behavioral bioassays. For example, diurnal rhythm affects the attractiveness of Dacini males to lures. In *Z*. *cucurbitae*, males were more attracted to cue-lure in the morning as compared to that in the mid-afternoon, while the attractiveness of cue-lure was lowest in the afternoon [60]. Similar observations have been reported for two ME-sensitive species, *B*. *opiliae* and *B*. *cacuminata* [61,62]. In *B. dorsalis* males, laboratory bioassays revealed that the attraction to ME occurred throughout the day until the onset of mating behavior at dusk, whereas field studies using synthetic ME demonstrated that peak attraction of males to the lure occurred in the morning [63]. After the male attraction, ME is ingested, biotransformed into sex pheromonal components, and subsequently transported from the crop to the rectal gland through the haemolymph in ME-sensitive species. [24,64,65,66,67]. This process may require time to ensure sufficient accumulation and release of the pheromones [65]. Thus, acquisition of ME in the morning may be advantageous for ME-fed males to mate at dusk [31].

Attraction of certain Dacini male flies to lures is known to be age-dependent, which is correlated with sexual maturity in ME-sensitive species, such as *B*. *dorsalis*, *B*. *carambolae*, *B*. *opiliae*, and *B*. *umbrosa*, and RK-sensitive species, such as *Z*. *cucurbitae* [62,68,69,70,71,72,73]. However, attraction of male flies to the lure, zingerone, also occurred before sexual maturation in the Jarvis’s fruit fly *B*. *jarvisi*, under normal laboratory breeding conditions, suggesting this species has an olfactory system that could respond to zingerone even at immature stages [74].

Photoperiodicity involving light intensity is a crucial factor that affects mating in the tropics at consistent temperatures [75]. At dusk, concomitant with decreased light intensity below 100 lx, mating entailed male intrinsic behavioral patterns, such as short bursts of wing fanning (also termed pheromone-calling) to release male pheromone compounds from the rectal gland through the anus [76]. Furthermore, light intensity has been shown to modulate female mating and oviposition affected by contact and/or volatile fruit stimuli in *B*. *oleae* [77]. Following mating, female tephritid flies have been shown to behaviorally switch their preference for host fruit odors rather than pheromones of conspecific males [78]. The olfactory-mediated behavior in females is modulated by male accessory gland fluid [78]. Additionally, nutritional status can also modulate the odor preference of mature females. In *B*. *dorsalis*, protein-fed females were more attracted to host fruit odors than to protein odors, regardless of their mating status [79]. In contrast, protein-deprived mated females were equally attracted to the fruit and protein odors [79].

Nutritional status also affects the mating behavior of males. Adult Dacini flies are usually provided with a diet consisting of a mixture of sugar and yeast (enzymatic hydrolysate) as a rich source of carbohydrates and proteins in laboratories [80]. Dietary protein strongly affected mating success in *B*. *dorsalis* and *B*. *tryoni* [81,82]. Food deprivation, even in short-term treatments (30 h), decreased mating success in *B*. *dorsalis* [83]. In *B*. *tryoni*, feeding a yeast hydrolysate-supplemented diet increased male attraction to cue-lure [84]. Conversely, protein deprivation reduced the attractiveness of males to cue-lure in *B*. *tryoni* males [84], perhaps due to delayed sexual maturation. It is noteworthy that the ingestion of male attractants also affects mating success. In *B*. *dorsalis*, ME ingestion enhanced wing-fanning (pheromone calling) behavior, thereby increasing pheromone release and attracting more females [85]. Even when providing a low-quality diet, ME-supplementation could enhance mating success in males [83]. For RK-sensitive species, ingestion of RK along with yeast hydrolysate also promoted sexual maturation and mating success in *B*. *tryoni* [86,87]. In contrast, supplementation with RK or cue-lure reduced attraction to cue-lure-baited traps in *B*. *tryoni* and *Z*. *cucurbitae* [88,89,90]. When newly emerged adults were ingested with RK, no significant effects on sexual maturation, mating success, and attraction to cue-lure-baited traps were observed in *Z*. *cucurbitae*, suggesting that different metabolic processes, including transport and accumulation of male attractants, distinctly affect the physiology and behavior among Dacini species [91].

As various factors associated with physiological status affect chemosensory-related behaviors in Dacini fruit flies as mentioned above, an evaluation of any behavioral changes associated with chemosensory perception requires careful conditioning. In field behavioral bioassays, the conditions of the bioassays should be similar to the field conditions of the wild population, lest the bioassays are improperly designed and interpreted [92]. In laboratory behavioral bioassays, we should be aware of the possible differences between laboratory and field strains of insects. In mass-rearing facilities, genetic changes in *Z*. *cucurbitae* related to developmental and behavioral traits have been observed in more than 40 successive generations [93,94]. Furthermore, various abiotic and biotic environmental factors, such as light intensity, temperature, and dietary conditions, must be acknowledged [58]. Therefore, appropriate behavioral bioassays focusing on the problems underlying experimental design are required to identify specific chemicals that influence the behavior of flies.

## 5. Major Molecular Components in Insect Chemoreception

Insect chemosensory receptors consist of three types of insect-specific superfamilies: olfactory receptors (ORs), gustatory receptors (GRs), and ionotropic receptors (IRs) [51,95,96,97,98]. In addition, odorant binding proteins (OBPs) are thought to facilitate the transport of hydrophobic ligands through the aqueous sensillum lymph to appropriate ORs [95,99,100]. The roles of these chemosensory components have been mainly characterized by studies on *D. melanogaster*, taking advantage of a rich repertoire of genetic tools. The chemosensory receptors are thought to form ligand-gated ion channels [101,102,103,104,105,106]. Among the chemosensory receptors, insects use ORs, IRs, and carbon dioxide receptors belonging to the GR family to detect olfactory signals. GRs and IRs mediate the gustatory signals. Thus, IRs function not only in olfaction, but also in gustation. Since the development of next-generation sequencing (NGS), numerous chemosensory-related genes have been reported or registered in insects, including Dacini fruit flies [42,107,108,109,110,111]. Furthermore, draft genome sequences of five Dacini fruit flies (Bioproject ID)—*B*. *dorsalis* (PRJNA273958), *B*. *latifrons* (PRJNA351211), *B*. *tryoni* (PRJNA241080), *B*. *oleae* (PRJNA293367), and *Z*. *cucurbitae* (PRJNA273817)—have been deciphered, and their reference sequences are now available on the NCBI website. Genetic information enables identification of sequences coding candidate chemosensory components.

### 5.1. OBPs

OBPs are small, water-soluble, extracellular proteins that are involved in the first step of insect chemosensory processing [100]. A member of the OBP family was first reported as a pheromone binding protein (PBP) in Lepidoptera [99]. OBPs are responsible for transporting external odorants that pass through the pores of the sensilla into ORs on the odorant receptor neurons, thereby initiating olfactory signal transduction. Nevertheless, there is controversy regarding whether ORs are activated by an odorant-OBP complex or a single odorant molecule. Currently, there are two hypotheses on the mode of action of OBPs [112,113]: 1) OBPs bind with odorants, then form a complex in which the allosteric complex activates the receptor; 2) OBPs bind with odorants to delay odorant release in the sensillum lymph. In the second hypothesis, OBPs have been suggested to maintain the stability of the insect olfactory system by buffering the effects of sudden changes caused by odorant stimuli, rather than activating the receptor, while the odorants can directly activate the ORs [113]. Notably, a recent study has shown that CRISPR/Cas9-mediated knockout of PBP genes in the tobacco cutworm *Spodoptera litura* reduced antennal and behavioral responses to sex pheromone components, but did not functionally abolish them [114], which supports the idea that OBPs act as mediators for the transport of hydrophobic ligands in the sensillum lymph.

ME and RK are powerful lure components that attract Dacini fruit flies. OBPs that participate in the perception of ME in fruit flies have been identified, although none have been characterized for RK perception. For instance, there are 49 OBPs annotated in *B*. *dorsalis* [115]. In this review, we describe the names of OBPs in *B*. *dorsalis* according to the nomenclature of *D*. *melanogaster* [115], and also indicate names used in previous papers (within parentheses). Among the OBPs, BdorOBP56f-2 (BdorOBP2) was found to be involved in the perception of ME, with significant ME-induced transcriptional expression in *B*. *dorsalis*. Furthermore, BdorOBP56f-2 gene silencing by RNAi reduced the sensitivity of *B*. *dorsalis* to ME significantly [116]. In addition to BdorOBP56f-2, BdorOBP83b (BdorOBP83a-2) also affected the sensitivity of *B*. *dorsalis* to ME [117].

Other than *Bactrocera* species, OBPs have also been characterized in Ceratitidini and Carpomyini fruit flies. (*E*,*E*)-α-Farnesene has been regarded as one of the pheromone components of the Mediterranean fruit fly *Ceratitis capitata* (Ceratitidini), which is released by mature males [118,119]. At present, 46 OBP genes have been identified in *Ce*. *capitata* [120]. Among these OBPs, CcapOBP22 (previously named CcapOBP69a) and CcapOBP83a-2 showed high affinity for (*E*,*E*)-α-farnesene [119,121]. In addition, CvesOBP5 and CvesOBP6 also showed strong binding capacity to nonanal, which is a pheromone component of the tephritid species, *Carpomya vesuviana* (Carpomyini) [122].

Recent studies have indicated that γ-octalactone, benzothiazole, ethyl tiglate, 1-octen-3-ol, and β-caryophyllene can attract gravid female Dacini flies for oviposition [10,41,123]. OBPs participating in the perception of these odorants have been progressively identified. It has been documented that CvesOBP1, CvesOBP4, and BminOBP9 identified from *B*. *minax* showed high affinity for β-caryophyllene in an in vitro binding assay [122,124]. Furthermore, BdorOBP84a-1, identified from *B*. *dorsalis*, also showed moderate affinity to several volatiles, including 1-octen-3-ol [117].

Additionally, several OBPs have been reported to have a strong affinity for several common host volatiles. Research on BminOBP9 (BminGOBP99a) of *B*. *minax* showed that it bound not only to β-caryophyllene but also to other citrus volatiles, such as ocimene and myrcene [124]. It was also observed that BdorOBP99b (BdorOBP99a) of *B*. *dorsalis* had a high affinity for host volatiles, such as ocimene and limonene [124]. CvesOBP2 of Ca. *vesuviana* specifically bound well with the (*Z*)-3-hexenyl acetate contained in the host fruit [122]. Characterized OBPs and their main ligands are summarized in Table 1 and Figure 2.

### 5.2. ORs

Among the insect chemosensory receptors, ORs have been relatively well characterized as heteromeric ligand-gated ion channels that consist of a specific OR and a highly conserved co-receptor ORCO [104,105,125]. Recently, high-resolution structural analysis using cryo-electron microscopy demonstrated that ORCO and OR subunits assemble into heterotetramers [101]. Since in vitro functional analyses using *Xenopus* oocytes and culture cell lines have been established [95], insect ORs including Dacini fruit flies have been characterized. Since a specific odorous ligand is tuned to a specific OR in the insect olfactory system, the identification of the ligands for uncharacterized ORs could provide essential information to understand how various chemicals in the external environment are recognized by insects. However, only a limited number of ORs have been functionally characterized in Dacini fruit flies, although extensive transcriptome data derived from chemosensory organs have been accumulated. Most recently, several ORs of *B*. *dorsalis* and *B*. *minax* that respond to semiochemicals, including ME and host plant volatiles, have been characterized, as mentioned later.

The identification of chemosensory receptors responsible for the detection of male attractants is a significant challenge. Among the three chemosensory receptor families, ORs have been relatively well analyzed in Dacini fruit flies. Sex-specific behaviors, such as male attraction or female oviposition, stimulated by plant semiochemicals are observed in the life cycles of fruit flies. Although statistical differences in the transcriptional levels of several ORs between the sexes have been reported [111,126], sex-specific expression of any OR has not been reported. This is different from the cases of lepidopteran sex pheromone receptors responding to female pheromones, all of which are specifically expressed in male antennae [127,128,129,130,131,132,133]. The ME receptor has been considered as an OR, because a previous study has reported that ORCO is required for ME detection in *B*. *dorsalis* [134]. In this study, ME-exposure enhanced the transcriptional level of *ORCO*. Furthermore, RNAi-mediated knockdown of *ORCO* reduced the attractiveness of ME in the treated males. Most importantly, a recent study characterized OR88a as the ME receptor [135]. This study showed that the transcriptional level of *OR88a* was enhanced by exposure to ME, as seen for *ORCO*. Crucially, in vitro functional analysis using the *Xenopus* oocyte system revealed an obvious response of oocytes co-expressing OR88a and ORCO to ME. RNAi-mediated knockdown of *OR88a* decreased the attraction of male flies to ME. Further experiments, such as knockout of this gene using genome editing, will provide clear evidence for OR88a as a bona fide ME-receptor. It is also important to identify chemosensory receptors responding to male sex pheromones, because male attractants are converted into sex pheromones in ME-sensitive fruit flies. As mentioned above, ME is oxidized into DMP and E-CF as male sex pheromones in *B*. *dorsalis*. The structures of DMP and E-CF are very similar to ME; therefore, it is intriguing whether ME-receptors or other homologous chemosensory receptor(s) respond to both sex pheromones with a broad spectrum. Otherwise, distinct chemosensory receptors may specifically respond to individual pheromones. It is also interesting to refer to the different patterns of pheromone components among the related species of *B*. *dorsalis*. The carambola fruit fly *B*. *carambolae*, which is very close to *B*. *dorsalis*, produces only E-CF from ME [136]. In contrast, the peach fruit fly *B*. *zonata* and *B*. *correcta* possess binary mixtures of ME-oxidates in an approximate ratio of 1:1 in the male rectal glands, as shown in *B*. *dorsalis*, but their components were different [43,137]. *B*. *zonata* accumulated DMP and (*Z*)-coniferyl alcohol (Z-CF), whereas *B*. *correcta* accumulated (*Z*)-3,4-dimethoxycinnamyl alcohol (Z-DMC) and Z-CF (Figure 3). The patterns in which the related species partially share a phenylpropanoid component, as both *B*. *dorsalis* and *B*. *zonata* contain DMP, raising a question regarding similarities of male sex pheromone receptors. Since the different binary patterns in the pheromone components might play a critical role in sexual isolation of related species, comparative analysis of pheromone receptors provides insight into evolutionary speciation driven by differentiated chemosensory receptors. It is also interesting to note the difference in informative processing between sexes to male attractants and sex pheromones.

Other than phenolic metabolites, various compounds also accumulate in male rectal glands. Although most components have not been functionally characterized, 1-nonanol analogs that accumulate in the rectal glands as major aliphatic components have shown semiochemical activities in some fruit flies. For example, 6-oxo-1-nonanol was detected in the male rectal glands of *B*. *carambolae*, a species closely related to *B*. *dorsalis*, together with a minor component, 1,6-nonanediol [138,139]. The amount of 6-oxo-1-nonanol increased concomitantly with sexual maturity and triggered a chemotactic behavior known as zigzag flight in conspecific females [140]. In *Z*. *cucurbitae*, 1,3-nonanediol was detected as one of the major components in the male rectal glands at a level similar to that of accumulated RK [141]. Although the defensive role of 1,3-nonanediol against a natural enemy has been demonstrated, its pheromonal role is unknown [23]. A recent study characterized the ORs of three Dacini fruit flies, *B*. *dorsalis*, *B*. *latifrons*, and *Z*. *cucurbitae*, responding to 1-nonanol analogs by focusing on OR74a homologs, because OR74a of *D*. *melanogaster* responds to 1-nonanol [142]. The three OR74a homologs responded to 1-nonanol, but their sensitivities were different. The OR74a homologs identified from *B*. *dorsalis* and *Z*. *cucurbitae* responded significantly to 6-oxo-1-nonanol, but not to 1,3-nonanediol, indicating similar binding properties of the homologous ORs.

Most recently, several ORs that respond to common plant volatiles have been characterized in *B*. *dorsalis* and *B*. *minax* by co-expression with their cognate ORCO using *Xenopus* oocytes. For *B*. *dorsalis*, BdorOR13a and BdorOR82a have been functionally characterized [126]. BdorOR13a specifically responds to 1-octen-3-ol among the 24 tested compounds, as reported in its homologs of *D*. *melanogaster* and the mosquito species *Anopheles gambiae* and *Aedes aegypti* [143,144,145]. BdorOR82a robustly responds to geranyl acetate, as reported in its homologs in *D*. *melanogaster* and *An*. *gambiae* [146,147]. Furthermore, BdorOR82a responds weakly, but significantly to linalyl acetate and farnesene isomers [126]. There are no reports examining whether OR82a homologs also respond to these terpenes in other insects. As the three terpenes have been detected in *T*. *catappa*, one of the hosts of *B*. *dorsalis* [8], it is intriguing whether BdorOR82a is associated with host-finding behavior. Indeed, geranyl acetate and ME elicited the strongest EAD responses from the antennae of *B*. *dorsalis* among the volatiles collected from *T*. *catappa* [8]. Linalyl acetate and farnesene isomers also elicited medial EAD responses [8]. In accordance with the antennal responses, the landing behavior of female *B*. *dorsalis* on spheres treated with geranyl acetate or farnesene isomers was observed in the laboratory experiments [126].

For *B*. *minax*, four ORs—BminOR3, BminOR12, BminOR16, and BminOR24—have been functionally characterized [148,149]. Of them, BminOR3, a homolog of OR13a, also showed a specific response to 1-octen-3-ol, as shown by BdorOR13a. Other ORs have been newly characterized in insect species. BminOR12 responds to eight various compounds, namely a linear aliphatic alcohol, three aliphatic esters, a monoterpenoid, an aromatic alcohol, an aromatic aldehyde, and an aromatic ester. BminOR12 responded to 1-octanol, but not to 1-nonanol or undecanol, suggesting that this receptor does not respond to aliphatic alcohols with chains longer than C-8. BminOR12 did not respond to the unsaturated C-8 alcohol, *cis*-3-hexen-1-ol. Thus, it would be interesting to know if the olefin moiety reduces the responsiveness of BminOR12. For aliphatic esters, BminOR12 responded to butyl acrylate, butyl propionate, and (*Z*)-3-hexenyl acetate, but not to ethyl acetate, ethyl butyrate, butyl butyrate, or isoamyl acetate. The structure-activity relationship between BminOR12 and its ligands will be clarified by further examining its response to various esters. For aromatic compounds, BminOR12 responds to benzyl alcohol, benzaldehyde, (*S*)-(+)-carvone, and methyl salicylate, but not to *p*-cymeme, 2-methoxy phenol, or ME. How BminOR12 distinguishes different compounds belonging to the same category must be addressed in the future. BminOR16 specifically responds to undecanol. The phylogenetic tree constructed from the ORs of five dipteran species, including *D*. *melanogaster*, revealed a unique clade comprising three *Bactrocera* ORs, namely BminOR16, BdorOR67c.2, and BdorOR67c.3 [149]. This clade does not contain an OR of other dipteran species, suggesting that the three ORs are probably specific to Dacini fruit flies.

BminOR24, a homolog of OR7a, responds robustly to linalool, but weakly to ME [148]. Notably, female flies were attracted to linalool, a component of their host citrus fruits [150]. Considering the robust response of BminOR24 to linalool, gravid females may search for their host fruits by recognition of linalool via this receptor. Multiple homologous OR7a genes diverged in Dacini fruit flies, although only one corresponding OR7a gene has been identified in *D*. *melanogaster* [111,126,142]. Therefore, understanding the functional roles and divergent processes of OR members belonging to the OR7a subfamily in Dacini fruit flies is interesting. The characterized ORs and their main ligands are summarized in Table 2 and Figure 2.

### 5.3. Other Chemosensory Receptors

Candidate GRs and IRs have been identified by transcriptome sequencing in several Dacini fruit flies [42,107,108,110,111], although their functional analysis has not been addressed. In vitro functional analysis of GRs and IRs has not been successful even in the model insect *D*. *melanogaster*, probably due to the complexity of receptor structures, with a few exceptions [106,151]. As male attractants trigger male feeding behaviors, it would be interesting if gustatory receptors are involved in the recognition of attractants. The tip-recording method [57] is one of the methods available for investigating whether gustatory sensilla are responsible for perceiving attractants. If chemosensory receptors are necessary for gustation, orphan receptors belonging to the GR and/or IR family could be candidates, because their roles as gustatory receptors have been mainly characterized in *D*. *melanogaster* [95,96,97]. As male attractants have a hydrophobic nature, binding proteins such as OBPs and CSPs might be necessary to dissolve them in an aqueous lymphatic fluid.

## 6. Future Research

### 6.1. Characterization of Chemosensory Receptors Responding to Male Attractants and Sex Pheromones

ORs are considered to be chemosensory receptors for male attractants in Dacini fruit flies. Indeed, OR88a has been characterized as a candidate ME receptor in *B*. *dorsalis* by heterologous expression using *Xenopus* oocytes, and behavioral bioassays of male flies treated with RNAi-mediated *OR88a* knockdown [135]. The involvement of the OR system in the chemoreception of ME has also been shown by RNAi-mediated *ORCO* knockdown in *B*. *dorsalis* [134]. These studies have shown that reduced expression of *OR88a* or *ORCO* leads to a decrease in the attraction of male flies to ME. However, a complete impairment of attractiveness was not observed in the male flies. Targeted mutagenesis using the CRISPR/Cas9 system is now available for *Bactrocera* species [152]. Therefore, the loss of function of specific chemosensory receptors of interest will clarify their roles in vivo. Though it is plausible that a specific OR responds to ME, studying IRs is also intriguing, because they have been characterized as molecular components required for both olfaction and gustation [97]. The requirement of multiple IRs to form a functional receptor complex has made it difficult to analyze their functional properties [102,103]. IRs expressed in sexually dimorphic taste neurons have been characterized as candidate pheromone receptors in *D*. *melanogaster* [153]. As male attractants in Dacini fruit flies have both odorant and taste properties, certain IR complexes may act as bifunctional receptors for both olfactory and gustatory functions. Since IRs are thought to form heteromeric complexes consisting of the co-receptor IR8a or/and IR25a ion channels [102], the genetic loss of function of IR8a and IR25a using the CRISPR/Cas9 system will provide an answer to this question.

As the responsiveness of male Dacini flies to their respective attractants are different among species [13], a comparative study of chemosensory receptors responsible for male attractants will provide insights into the structure-activity relationships from the point of view of the structural properties of chemosensory receptors. It will also unravel the evolutionary process of the distinct ME- and RK-sensitive fruit fly groups to adapt to their surrounding environment, such as the source of attractants derived from plants, including fruit fly orchids. As of the structural resemblance of ME and RK, the difference between the corresponding chemosensory receptors is interesting. A few differences in the critical amino acids in receptors possibly impart the distinct ligand-binding properties. Furthermore, both ME- and RK-sensitive fruit fly species are attracted to zingerone owing to its structural resemblance to ME and RK [23]. A recent study on the chemical modification of zingerone has shown the structural properties required for the attraction of *B*. *jarvisi* [154]. Therefore, it will be interesting to study partial structures in ME- and RK-receptors that are necessary for binding to zingerone. Notably, males of *B*. *latifrons* are attracted to and feed on α-ionone analogs [47,48,49,50]. The different skeletons of specific male attractants, i.e., α-ionone analogs for *B*. *latifrons* and aromatic compounds for *B*. *dorsalis* and *Z*. *cucurbitae*, suggest that chemosensory receptors in both these species bear greater similarity than those in *B*. *latifrons*. Nevertheless, the phylogenetic analysis of ORs shows that *B*. *dorsalis* and *B*. *latifrons* share more similar homologous genes than *Z*. *cucurbitae* [111,142], reflecting their phylogenetic relationships. Structural properties of chemosensory receptors responsible for the different affinities for the α-ionone analogs and aromatic compounds are important from various aspects, such as structure-activity relationships, evolutionary changes in chemoreception, and development of new lures.

Unlike male attractants, male sex pheromones released from the rectal glands as smoke affect females at a close distance [29,32]. Therefore, the nature of chemosensory receptors responsible for the detection of sex pheromones is of interest. ME-sensitive fruit flies utilize the analogous male sex pheromones derived from ME. Moreover, the related species share the same components partially (Figure 3) [43,136,137]. Therefore, slight differences in amino acids that are critical for ligand binding in chemosensory receptors may change the ligand-binding properties among the related species. In this case, the mode of acquisition for different amino acid sequences during the evolution of pheromone production and reception is very important. Although males accumulate ingested RK in their rectal glands, the role of RK is unclear [14]. Identification of chemosensory receptors for RK, and the subsequent loss of function analysis could provide a clue to understanding the roles of RK in sexual communication of fruit flies.

Various components other than the aromatic compounds such as ME and RK have been identified in the rectal glands. The most interesting case is the spiroketal, olean, in *B*. *oleae* [35,36,37]. The different responses to the enantiomers between males and females raise the question of how chiral discrimination is achieved. It is plausible that different, but similar, chemosensory receptors are responsible for sensing the (*R*)- and (*S*)-enantiomers. If so, the structural properties for enantiomeric specificity must be key for the different recognition between sexes. Other than spiroketals, various acetamides, cyclolactones, and aliphatic compounds have been identified [38], but their biological roles remain unknown. We may be able to presume the roles of the rectal gland components by characterizing chemosensory receptors responding to the uncharacterized components.

### 6.2. Characterization of Chemosensory Receptors Responding to Common Plant Volatiles

Recently, ORs responding to common plant volatiles have been functionally characterized in *B*. *dorsalis* and *B*. *minax* [126,148,149]. The results provide knowledge of the properties of ORs which may narrowly or broadly tune to plant volatiles. Among the common plant volatiles, the sesquiterpenoid β-caryophyllene attracts males of *B*. *correcta*, but not those of *B*. *dorsalis* [44]. As β-caryophyllene affects phytophagous insects in many aspects of plant-insect interactions, identification of chemosensory receptor(s) will provide information about the common recognition of plants via chemoreception in phytophagous insects.

### 6.3. Information Processing of Chemosensory Inputs

It is important to understand how chemosensory information is processed in the brain, because olfactory and gustatory inputs seem to be associated with each other in Dacini fruit flies. Olfactory neurons within the chemosensilla project their axons into the glomeruli within the antennal lobe of the brain in insects [51,155]. The glomeruli are the first relay sites where olfactory neurons are synaptically connected with the projection neurons conveying processed olfactory information to higher brain centers. Individual glomeruli receive convergent input from multiple olfactory neurons expressing the same ORs, indicating a functional module reflecting the nature of odor stimuli [156,157]. Thus, a comprehensive understanding of the neural mechanisms of odor detection and processing requires an olfactory sensory map of the brain. Recently, glomerular organization in the antennal lobe has been described in *B*. *dorsalis* [158]. In total, 64–65 glomeruli were anatomically classified in both sexes using synaptic antibody staining. The number of glomeruli was in the range of 50–70, in other dipteran species. By comparing male and female antennal lobes, eight glomeruli showing enlargement in males or females were identified [158]. As only males are strongly attracted to ME, it is of interest to know whether these sex-dimorphic glomeruli are involved in the recognition of ME. A recent study used Ca^2+^ imaging in *B*. *tryoni* to identify a region in the antennal lobe, which is the primary olfactory center in the brain responsible for the recognition of odors, including cue-lure [53]. The standard reference stimulus, octanol, elicited a distinct enhancement of Ca^2+^ concentration in the glomeruli of both sexes. It is expected that maxillary palp-innervated glomeruli are activated by cue-lure stimulation, because the maxillary palps, but not the antennae, respond to cue-lure in electrophysiological analyses. However, no evident response in the antennal lobe, including the glomeruli, was observed by cue-lure stimulation, probably due to technical difficulties in tracing projection from the maxillary palps [53].

In addition to the attractants, information processing of male sex pheromones in females of ME-sensitive fruit flies is also a crucial issue. Due to the structural similarities between ME and sex pheromones, identification of receptors and characterization of their neural projection patterns will help us better understand the mechanisms underlying the different responses and information processing of similar compounds between males and females. Similarly, differences in information processing of the enantiomers of the spiroketal, olean, between sexes in *B*. *oleae* is intriguing. It must be explored whether chemosensory receptor neurons discriminating the enantiomers project into similar or different area in the brain between males and females.

It is possible that ORs modulate male feeding behavior. In the blowfly *P. regina*, axonal projections of olfactory receptor neurons from the maxillary palps to the primary gustatory center, the subesophageal ganglion (SOG), as well as to the antennal lobes have been shown by an anterograde staining method [56]. Furthermore, overlapping projection of maxillary olfactory receptor neurons and labellar gustatory neurons in the SOG was observed, suggesting an interaction between olfactory input from the maxillary palps and gustatory stimuli in the labellum. In behavioral bioassays, an attractive odor, 1-octen-3-ol, increased the response of the proboscis extension reflex to sucrose via the maxillary palps, but not via the antennae, which supports the proposed integration of olfactory and gustatory information in the SOG. This framework may be applicable to attractant-feeding behavior in Dacini fruit flies, as voracious feeding behavior is observed after attraction to lures. In the mosquito, *An*. *gambiae*, the innervation of *ORCO*-expressing neurons from the proboscis into the subesophageal zone was observed [159]. However, no innervation of *ORCO*-expressing neurons from the olfactory organs, antennae or maxillary palps, was detected in this species. Thus, neural connections seem to differ among dipteran species. Comprehensive studies on the relationships between neural networks and behavioral outputs would provide insights into the mechanisms underlying information processing in multiple chemosensory systems.

### 6.4. Mechanisms of Physiological Change Associated with Mating Behavior

One of the reasons why Dacini males voraciously feed on attractants is to elevate mating success. Ingestion of male attractants activates the characteristic pheromone calling behavior, wing-fanning, and simultaneously disperses pheromones from their rectal glands [31,83,85,86,87,90]. Activation of male behavior by ingestion of attractants is an intriguing question from the point of view of insect endocrinology. In the Caribbean fruit fly *Anastrepha suspensa* (Toxotrypanini), treatment with juvenile hormone promoted male sexual maturation; therefore, males produced sex pheromones at earlier ages than untreated males [160]. Similarly, treatment with a juvenile hormone mimic, methoprene, accelerated sexual maturation and enhanced mating behavior of males in RK-sensitive *Z*. *cucurbitae* and *B*. tryoni [161,162]. Taken together, the accelerated sexual maturation by male attractants is possibly associated with the action of juvenile hormone. In contrast, no significant effect of methoprene on male sexual activity was observed in *B*. *dorsalis* or *Ce*. *capitata* [163]. The different effects on sexual maturation may be caused by different sensitivities to or the mode of action of juvenile hormone among fruit fly species. Recently, the molecular action of juvenile hormone, including binding of ligand to receptors and subsequent signaling pathways, has been well elucidated from studies on various insects, such as dipteran, lepidopteran, coleopteran, and hemipteran species [164,165]. This information will provide cues to elucidate the mechanisms underlying the sexual maturation-promoting action of male attractants and the possible involvement of juvenile hormone in Dacini fruit flies.

Recent studies have shown that the gut microbiome can modulate various aspects of physiology and behavior, such as reproduction and feeding, in Dacini fruit flies. Symbiotic bacteria affect the foraging behavior of adult flies towards nutritional resources in *B. dorsalis* [166,167], and oviposition behavior in *B*. *oleae* [168]. Considering that chemical stimuli are a crucial cue for locating appropriate food sources and oviposition sites, it is intriguing whether chemosensory perception is affected by the gut microbiome in Dacini fruit flies.

### 6.5. New Approaches Such as Promoter Analysis and Ectopic Expression Using New Genetic Techniques

Insect chemosensory mechanisms have been elucidated in *D*. *melanogaster* taking advantage of the availability of genetic techniques. For example, visualization of chemosensory neurons using fluorescent proteins linked to the promoter region of a target gene has revealed neuronal projections into an information processing region in the brain [155]. Ectopic expression of calcium indicators, such as GCaMP, in a target chemosensory neuron has enabled the monitoring of neuronal activities stimulated by ligands. Although these techniques are limited to *D*. *melanogaster*, the utilization of a binary expression system, the QF/QUAS system [169], in *Anopheles* mosquitoes has successfully provided insights into olfactory innervations in the brain and olfactory system underlying the mode of action for insect repellents [159,170]. The successful generation of transgenic strains for the QF/QUAS system has enabled robust expression of reporters in chemosensory neurons. The labeling of *ORCO*-expressing neurons with GFP represents detailed anatomical maps of olfactory innervations in the brain [159]. The ectopic expression of GCaMP in the *ORCO*-expressing neurons allowed the direct visualization of olfactory responses to odorants in olfactory neurons, which demonstrated the different modes of action between natural repellents and synthetic repellents [170]. Furthermore, the advent of genome editing has made transgenic techniques applicable to a wide range of non-model insects [171]. For example, a CRISPR/Cas9-based knock-in technique has been developed to elucidate the roles of an ion channel, *ppk301*, in the control of egg-laying initiation and choice in the mosquito, *Ae*. *aegypti* [172]. The generation of a transgenic reporter strain in which a transcriptional activator was inserted into the *ppk301* locus led to a loss-of-function mutation. This also enabled ectopic expression of the transcriptional activator in *ppk301*-expressing cells. GFP expression using this *ppk301*-activator revealed projections of *ppk301*-expressing sensory neurons into central gustatory centers, the subesophageal zone, in the brain. Furthermore, the genetically introduced calcium sensor GCaMP6s in *ppk301*-expressing sensory neurons showed obvious responses to water using calcium imaging.

Thus, useful genetic tools developed in *Drosophila* could also be adopted for similar uses to elucidate chemosensory mechanisms in Dacini flies, as shown by the mosquito studies. A visualization of the innervation patterns of *ORCO*- and *IR*-expressing neurons provides information on chemosensory processing in the brain. Generation of transgenic strains, in which a transcriptional activator is regulated by a promoter of chemosensory receptors, render various analyses possible. For example, ectopic expression of GCaMP in cells of a chemosensory receptor would enable the identification of ligands for the receptor. These new approaches to study Dacini flies will not only broaden our knowledge of the chemosensory mechanisms underlying evolutionary adaptation to environmental changes, but also provide novel techniques that are useful for pest management.

## 7. Conclusions

Herein, we have reviewed chemoreception and related topics in Dacini fruit flies from various aspects of chemical ecology, physiology, neuroscience, and molecular biology. Due to the importance of pest management and the intriguing life cycle of Dacini fruit flies, a large variety of semiochemicals intimately associated with the life cycles of various species have been well identified since the discovery of a male lure for Dacini fruit flies in 1912 [16]. In addition, only a few molecular components required for chemoreception have been characterized for a limited number of species. In particular, identification of ligands for ORs has been achieved in the past few years. Molecular and cellular understanding of chemosensory mechanisms is much more advanced in the model insect *D*. *melanogaster* owing to the availability of various genetic tools. Currently, several genetic tools developed in *D*. *melanogaster* have become available for non-model species using CRISPR/Cas9 techniques. Thus, the mechanisms underlying the perception by peripheral organs and information processing in the brain may be elucidated in Dacini fruit flies by utilizing these new tools. Numerous studies on the unique behavioral and physiological traits associated with chemosensory perception have been conducted in Dacini fruit flies; therefore, understanding the molecular and physiological mechanisms of the chemosensory system provides insights into how Dacini fruit flies have optimally adapted to their surrounding environments via chemoreception at the molecular level. Such detailed understanding of chemical communication strategies also provides cues for the development of new biocontrol methods.

## Figures and Tables

**Figure 1 insects-12-00106-f001:**
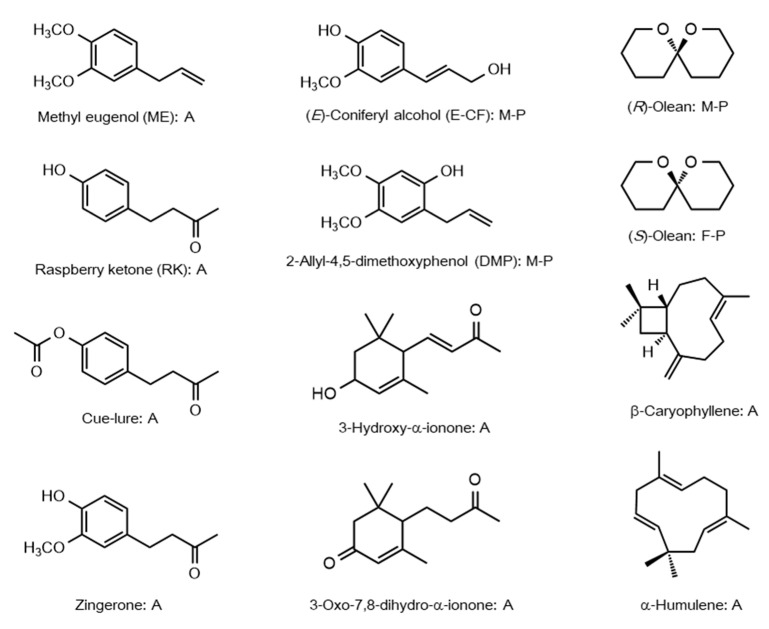
Structures of representative attractants and sex pheromones in Dacini fruit flies. A: Attractant; M-P: Male sex pheromone; F-P: Female sex pheromone.

**Figure 2 insects-12-00106-f002:**
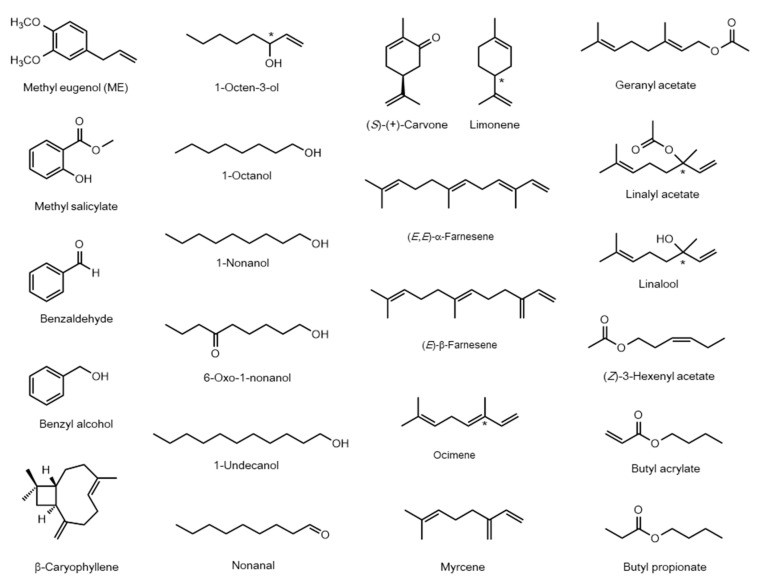
Structures of ligands for odorant biding proteins and olfactory receptors presented in Table 1 and Table 2. Stereocenters are shown by asterisks.

**Figure 3 insects-12-00106-f003:**
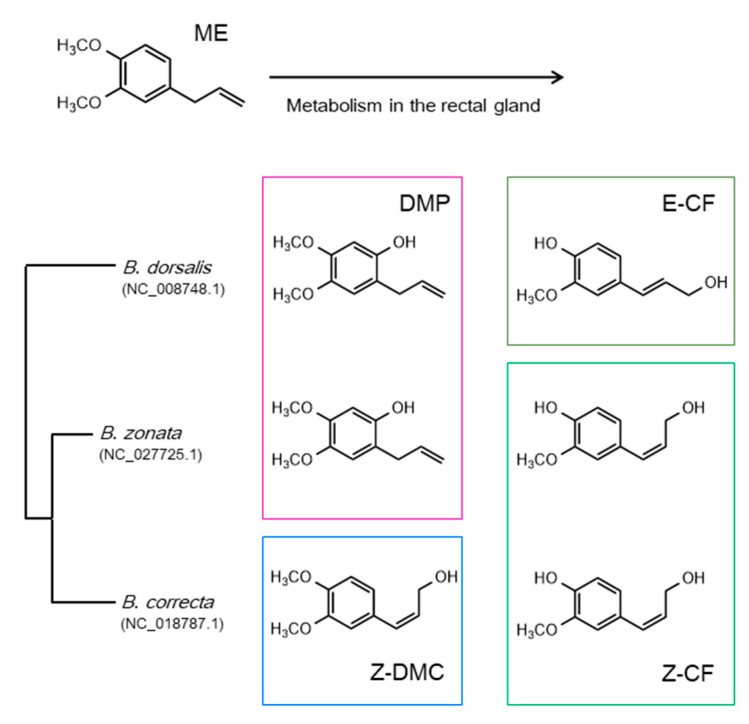
Metabolites of methyl eugenol in three *Bactrocera*-related species. The phylogenetic tree was constructed from mtDNA sequences by the maximum likelihood method. The accession numbers of the mtDNA sequences are indicated in the parentheses. ME: methyl eugenol; DMP: 2-allyl-4,5-dimethoxyphenol; Z-DMC: (*Z*)-3,4-dimethoxycinnamyl alcohol; E-CF: (*E*)-coniferyl alcohol; Z-CF: (*Z*)-coniferyl alcohol (Z-CF).

**Table 1 insects-12-00106-t001:** OBPs and their high affinity odorants.

Species	Name	Accession Number	Ligand	Ref.
*Bactrocera dorsalis*	BdorOBP56f-2 (BdorOBP2)	KC559113	Methyl eugenol	[112]
	BdorOBP83b (BdorOBP83a-2)	KP743700	Methyl eugenol	[113]
	BdorOBP99b (BdorOBP99a)	KP743706	Limonene	[120]
			Ocimene	
	BdorOBP84a-1	MT474623	1-Octen-3-ol	[113]
*Bactrocera minax*	BminOBP9 (BminGOBP99a)	KY463449	β-Caryophyllene	[120]
			Ocimene	
			Myrcene	
*Ceratitis capitata*	CcapOBP22 (CcapOBP69a)	NP_001295335.1	(*E*,*E*)-α-Farnesene	[117]
	CcapOBP83a-2	XM_004523387.1	(*E*,*E*)-α-Farnesene	[115]
*Carpomya vesuviana*	CvesOBP1	KU975053	β-Caryophyllene	[118]
	CvesOBP2	KU975054	(*Z*)-3-Hexenyl acetate	[118]
	CvesOBP3	KU975055	α-Farnesene	[118]
	CvesOBP4	KX059394	β-Caryophyllene	[118]
	CvesOBP5	KU975056	Nonanal	[118]
	CvesOBP6	KX059395	Nonanal	[118]

**Table 2 insects-12-00106-t002:** Olfactory receptors that respond to semiochemicals in Dacini fruit flies.

Species	Name	Accession Number	*Drosophila* Homolog ^(1)^	E-Value	Ligand ^(2)^	Ref.
*Batrocera dorsalis*	BdorOR13a	FX985890	OR13a	3e−122	1-Octen-3-ol (+++)	[122]
*Batrocera dorsalis*	BdorOR63a-1	KP743726	OR63a	1e−95	Methyl eugenol (+)	[131]
*Batrocera dorsalis*	BdorOR74a	FX985927	OR74a	5e−63	1-Nonanol (+++)	[138]
					6-Oxo-1-nonanol (++)	
*Batrocera dorsalis*	BdorOR82a	FX985928	OR82a	2e−83	Geranyl acetate (+++)	[122]
					Farnesenes ^(3)^ (+)	
					Linalyl acetate (++)	
*Batrocera dorsalis*	BdorOR88a	KP743732	OR88a	1e−35	Methyl eugenol (+++)	[131]
*Batrocera latifrons*	BlatOR74a	FX986103	OR74a	2e−63	1-Nonanol (++)	[138]
*Bactrocera minax*	BminOR3 ^(4)^	MN537976	OR13a	5e−119	1-Octen-3-ol (+++)	[145]
*Bactrocera minax*	BminOR12	MN855530	OR63a	4e−109	Methyl salicylate (+++)	[145]
					Benzaldehyde (++)	
					(*Z*)-3-Hexenyl acetate (++)	
					Butyl acrylate (++)	
					Butyl propionate (++)	
					1-Octanol (+)	
					(*S*)-(+)-Carvone (+)	
					Benzyl alcohol (+)	
*Bactrocera minax*	BminOR16	MN537977	OR67c	9e−33	1-Undecanol (++)	[145]
*Bactrocera minax*	BminOR24	MN537978	OR7a	1e−50	Linalool (+++)	[144]
					Methyl eugenol (+)	
*Zeugodacus cucurbitae*	ZcucOR74a	FX986226	OR74a	7e−64	1-Nonanol (+++)	[138]
					6-Oxo-1-nonanol (+)	

^(1)^ Homolog of *Drosophila melanogaster* was searched using the flybase (http://flybase.org). ^(2)^ Responses are indicated in the parentheses as the following currents responded to ligands at 100 µM. +++: ≥100 nA; ++: ≥50 nA; +: <50 nA. ^(3)^ Two typical natural products are shown in Figure 2. ^(4)^ This OR is registered as BminOR5 in the NCBI website.

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
