# Peer review of "Recent Advancements in Studies on Chemosensory Mechanisms Underlying Detection of Semiochemicals in Dacini Fruit Flies of Economic Importance (Diptera: Tephritidae)"

_insects, 2021, doi:10.3390/insects12020106_

Round 1
Reviewer 1 Report
The authors of this review describe recent studies on Chemoreception of semiochemicals in Dacini fruit flies. Reviewing the literature requires an enormous amount of work and the manuscript is interesting, still, there are several issues that the authors may wish to consider in revising this paper. The review needs major restructuring and minor revision prior to acceptance in the Insects journal.
The general issues that need to be addressed are the following:
-
- The tribe Dacini with more than 930 species includes three crucial genera: Bactrocera, Zeugodacus and Dacus, so a well-defined title or a rewriting introduction is necessary.
- It would be better for the reader to have a clear definition of the terms used, since if we speak about “plant volatiles" it is a very generic term that does not imply a chemical message. The plant semiochemical term would perhaps be more useful.
- Page 2. 2.1 (line 77 to 123). This section has to be completely rewritten. From Howlett’s (1912 and 1915) publications about the 3-4 dimethoxy-1 allylbenzene (ME) attractively to male Z. diversus an B. dorsalis many male-lures has been identified. Methyl eugenol widely occurs in many plant parts of a number of plant specie, so therefore, whether or not the species is polyphagous is an important issue that should be included. Further Drew and Hooper (1981) reported that each Dacini species responded only to one of these attractants and some species did not respond to
- Figure 1. It would be helpful if the semiochemical effect (sex pheromone, plant kairomone, para-kairomone,…..) of the chemical structures were shown in the figure.
- Paragraph 2.2 8line 124 to 160). This paragraph has several problems and it is very difficult to read and follow. Activity of different volatiles (sesqui and norsesquiterpenes) is interspersed in different bioassays (field test, EAG, GC-EAD, wind tunnel assay,….) with different species and sexes, a real puzzle. Please, it is necessary to rewrite it and perhaps with a support table it would be easier to follow. Regarding a-ionol and derivatives was McGover et. al.(1989) who indicated its attractive activity against latifrons males.
- Figure 2. Please remove structures that already appear in figure 1, they are redundant. Indicate in structures with stereocenters the absolute configuration (easily accessible, for example on the pubchem website). Any case, please, remove “(an isomer)”.
- Pag 5. Line 199. The title refers "chemosensory", but there is no information on chemoreception in the content. Please consider modifying the paragraph title.
Author Response
Response to Reviewer 1
We appreciate your comments and suggestions. We provide a point-by-point response to the comments, as below.
- The tribe Dacini with more than 930 species includes three crucial genera: Bactrocera, Zeugodacus and Dacus, so a well-defined title or a rewriting introduction is necessary.
Response: We describe the Tribe Dacini more accurately in line 56-58, citing the reference by Krosh et al., 2012. We also changed the title.
- It would be better for the reader to have a clear definition of the terms used, since if we speak about “plant volatiles" it is a very generic term that does not imply a chemical message. The plant semiochemical term would perhaps be more useful.
Response: We leave them as plant volatiles, because not all have yet been characterized in their functions, and to use the term plant semiochemicals is inaccurate
- Page 2. 2.1 (line 77 to 123). This section has to be completely rewritten. From Howlett’s (1912 and 1915) publications about the 3-4 dimethoxy-1 allylbenzene (ME) attractively to male Z. diversus an B. dorsalis many male-lures has been identified. Methyl eugenol widely occurs in many plant parts of a number of plant specie, so therefore, whether or not the species is polyphagous is an important issue that should be included. Further Drew and Hooper (1981) reported that each Dacini species responded only to one of these attractants and some species did not respond to
Response: 1) We introduce the first discovery by Howlett, and identification of male-lures since then (line 82-84). 2) We describe the distribution of ME in many plant parts of a number of plant species in Introduction (line 64-65). 3) It is interesting issue whether or not the ME-sensitive species is polyphagous. Indeed, Bactrocera dorsalis is highly polyphagous with over 120 host plants across 42 families. However, only males, but not host-finding females, are strongly attracted to ME. Therefore, we cannot discuss about the association of diet breadth with ME-distribution. In this review about chemosensory mechanisms, we do not deal with the topic. 4) We revised the sentence in line 86-87, citing Drew and Hooper, 1981.
- Figure 1. It would be helpful if the semiochemical effect (sex pheromone, plant kairomone, para-kairomone,…..) of the chemical structures were shown in the figure.
Response: We show the categories, A: Attractant; M-P: Male sex pheromone; F-P: Female sex pheromone, in Fig. 1.
- Paragraph 2.2 8line 124 to 160). This paragraph has several problems and it is very difficult to read and follow. Activity of different volatiles (sesqui and norsesquiterpenes) is interspersed in different bioassays (field test, EAG, GC-EAD, wind tunnel assay,….) with different species and sexes, a real puzzle. Please, it is necessary to rewrite it and perhaps with a support table it would be easier to follow. Regarding a-ionol and derivatives was McGover et. al.(1989) who indicated its attractive activity against latifrons males.
Response: 1) We have arranged this paragraph according to major Dacini species and plant volatiles. We do not have any idea to improve the paragraph in another way. We believe that it is not necessary to rewrite the entire paragraph. Because volatiles are described as individual compounds, categories, or plant odors, according to individual cases, it is impossible to make a support table. 2) We have already cited the discovery by McGovern et al. (Flath et al., 1994). “McGover et. al. (1989)” is a US patent, but not a scientific paper, therefore, we do not cite it as a reference.
- Figure 2. Please remove structures that already appear in figure 1, they are redundant. Indicate in structures with stereocenters the absolute configuration (easily accessible, for example on the pubchem website). Any case, please, remove “(an isomer)”.
Response: Although some structures are duplicated in Figs. 1 and 2, these compounds are shown in different meanings in the figures. Therefore, we leave them in Fig. 2. We indicate the asterisks at stereocenters, and remove “(an isomer)”.
- Pag 5. Line 199. The title refers "chemosensory", but there is no information on chemoreception in the content. Please consider modifying the paragraph title.
Response: We changed it to “Chemosensory-Related Behaviors Modulated by Physiological Status” (line 206).

Reviewer 2 Report
I find the manuscript by Ono et al. very informative and comprehensive and have no major issues with this review. Here are the points that need to be addressed:
- line 52: provide full taxonomy for Tephritid flies
- lines 62-63: Are there any other natural occurrences of ME besides fruit fly orchids? If so, please list these.
- line 114: I suggest to leave out "There are two (R)- and (S)-enantiomers in olean", because it is misleading, or correct to `the (R) and (S) enantiomers of olean...`.
- line 139: What do you mean by "behaviors in citrus fruits"? Please clarify.
- line 164: `various` - vague, be more specific
- lines 203-205: Sentence is not clear - Are you comparing things by using `than`?
- line 209: I suggest to use `biosynthesized` instead of `biotransformed`.
- line 218: delete `equipped` from sentence
- line 290: correct to `CRISPR/Cas9`
- Figure 2: Leave out `an isomer` after certain compound names, because in Tables 1 and 2 no enantiomers are specified. The structures correctly represent the way compounds are mentioned in tables.
- line 506: What do you mean by `smoke`? Do you mean `plume`?
- line 513: delete `a` from `is a very important`
Author Response
Response to Reviewer 2
We appreciate your favorable comments. We provide a point-by-point response to the comments, as below.
line 52: provide full taxonomy for Tephritid flies
Response: We provide it as “Diptera: Tephritidae”.
lines 62-63: Are there any other natural occurrences of ME besides fruit fly orchids? If so, please list these.
Response: We describe the distribution of ME in Introduction (line 64-65).
line 114: I suggest to leave out "There are two (R)- and (S)-enantiomers in olean", because it is misleading, or correct to `the (R) and (S) enantiomers of olean...`.
Response: We revised it (line 121).
line 139: What do you mean by "behaviors in citrus fruits"? Please clarify.
Response: We changed it to “in citrus fruit flies” (line 146).
line 164: `various` - vague, be more specific
Response: We don't think it is vague, because the antennae are known to be the major olfactory organs in insects to detect myriads of compounds.
lines 203-205: Sentence is not clear - Are you comparing thingsby using `than`?
Response: We revised the sentence to be clear (line 211-212).
line 209: I suggest to use `biosynthesized` instead of`biotransformed`.
Response: Because ME-metabolites that function as male sex pheromonal compounds are directly derived from ME (Hee and Tan, 2004, 2005, 2006), it will be more appropriate to use the term biotransformed.
line 218: delete `equipped` from sentence
Response: We deleted it (line 225).
line 290: correct to `CRISPR/Cas9`
Response: We revised it (line 297).
Figure 2: Leave out `an isomer` after certain compound names, because in Tables 1 and 2 no enantiomers are specified. The structures correctly represent the way compounds are mentioned in tables.
Response: We removed “an isomer”. Because some previous studies have not indicated steric structures, we provide structural formulas, just indicating stereocenters.
line 506: What do you mean by `smoke`? Do you mean `plume`?
Response: It should be smoke as described by the authors in their laboratory cage assay. Please see the references of Koyabashi et al. (1978) and Ohinata et al. (1982).
line 513: delete `a` from `is a very important`
Response: We deleted it (line 519).
